# Sculpting Features from Noise: Reward-Guided Hierarchical Diffusion for Task-Optimal Feature Transformation

**Nanxu Gong**[1][*] **Zijun Li**[2][*] **Sixun Dong**[1]**, Haoyue Bai**[1]**, Wangyang Ying**[1],
**Xinyuan Wang**[1]**, Yanjie Fu**[1][†]
[1]Arizona State University
[2]National University of Singapore,
nanxugong@outlook.com, zijunli@u.nus.edu,
{sixundong, haoyuebai, wying4, xwang735, yanjie.fu}@asu.edu,

## Abstract

Feature Transformation (FT) crafts new features from original ones via mathematical operations to enhance dataset expressiveness for downstream models. However, existing FT methods exhibit critical limitations: discrete search struggles with enormous combinatorial spaces, impeding practical use; and continuous search, being highly sensitive to initialization and step sizes, often becomes trapped in local optima, restricting global exploration. To overcome these limitations, DIFFT redefines FT as a reward-guided generative task. It first learns a compact and expressive latent space for feature sets using a Variational Auto-Encoder (VAE). A Latent Diffusion Model (LDM) then navigates this space to generate high-quality feature embeddings, its trajectory guided by a performance evaluator towards task-specific optima. This synthesis of global distribution learning (from LDM) and targeted optimization (reward guidance) produces potent embeddings, which a novel semi-autoregressive decoder efficiently converts into structured, discrete features, preserving intra-feature dependencies while allowing parallel inter-feature generation. Extensive experiments on 14 benchmark datasets show DIFFT consistently outperforms state-of-the-art baselines in predictive accuracy and robustness, with significantly lower training and inference times. Our code and data are publicly available at `https://github.com/NanxuGong/DIFFT`

## 1 Introduction

In real-world practices, raw data features often contain imperfections and complex interdependencies that hinder model performance. To address the challenges, Feature Transformation (FT) has been widely used, because it can reconstruct distance measures, reshape discriminative patterns, and enhance data AI readiness (structural, predictive, interaction, and expression levels). Given a raw feature set $\mathcal{F}_{\text{raw}} = [f_1, f_2, \ldots, f_n]$ and an operator set $\mathcal{O}$ (e.g., $\log, +, *$), feature transformation aims to apply mathematical operations to original features and construct a better feature set, for instance, $[f_1 * f_2, \log f_3, f_4 + f_5]$. In this paper, we study the FT problem that aims to apply mathematical operations to cross original features and create a better feature set.

In prior literature, manual transformation heavily relies on domain knowledge with limited generalization. Discrete search methods explore feature-operator combinations using reinforcement learning, evolutionary algorithms, and genetic programming [6, 29, 26, 7]. However, these approaches suffer

---

[*]Equal contribution
[†]Corresponding author

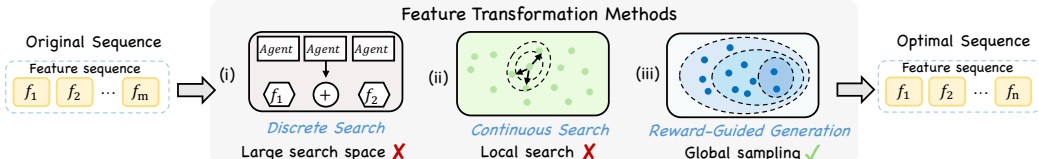

Figure 1: Motivation example. Discrete search methods explore various feature combinations directly, but are often challenged by the sheer scale of the resulting combinatorial space. Continuous search methods, on the other hand, iteratively refine solutions from initial or current points, yet frequently converge to local optima. In contrast, our proposed reward-guided generation paradigm, by leveraging the global sampling ability of diffusion models, aims to discover solutions closer to the global optimum.

from the large combinatorial discrete search space. With the rise of GenAI and the concept of any modality as language tokens, researchers see feature sets as token sequences. For instance, a transformed feature set $[f_1 * f_2, \ log f_3, \ f_4 + f_5]$ can be represented by a postfix expression: "$f_1 \ f_2 \ *$, $f_3 \ \log, \ f_4 \ f_5 \ +$" to reduce sequence length and ambiguity. FT therefore is reduced into a GenAI task: learning to generate a task-optimal feature token sequence given tasking data. Existing solutions follow an embedding-search-generation paradigm [27, 31, 28]: i) learning feature token sequence embeddings; ii) gradient ascent of optimal feature set embeddings; iii) autoregressive decoding of identified embeddings into transformed tables. The key idea is gradient search of optimal feature set embedding in a continuous embedding space, instead of search in the large discrete space.

Although the continuous search paradigm of generative FT introduces computational benefits, there are two major challenges. First, the continuous gradient search can get stuck in a local optima and is sensitive to initialization and search boundaries, leading to low robustness. A question that triggers our curiosity is: how can we better learn a generalized representation space describing all high or low-performance feature transformations on tasking data? Second, autoregressive decoding assumes that two consecutive features in a sequence are dependent, and, thus, considers redundant contexts and long attentions and is time-costly. For instance, while the tokens in one transformed feature $f_1 * f_3$ are interdependent, $f_1 * f_3$ is not dependent on another transformed feature $log(f_3)$. Another interesting question is: how can we enhance generative efficiency without compromising feature transformation embedding decoding quality? Thus, we need a new learning paradigm that goes beyond the continuous search paradigm in a static embedding space.

**The Opportunity.** As one of the generative frameworks, diffusion models learn the data distribution by training on a forward noising process that incrementally adds noise to input data and a reverse denoising process that reconstructs the data step-by-step, in order to capture the probability landscape of the data. Once trained, the reverse diffusion process can be guided—either conditionally or through optimization cues—to sample the most probable or desirable data points from the learned distribution in a generative fashion. This offers a compelling alternative to continuous gradient search.

**Our Insights: an integrated reward-guided diffusion and hierarchical decoding paradigm.** We formulate generative FT as a diffusion generation task, instead of a gradient search task. To overcome the limitations of gradient search, our first insight is optimization as reward-guided diffusion generation. The idea is to leverage forward diffusion and reverse diffusion to learn the global and generalizable distribution of feature set embedding space, then leverage reward (gradient from a performance evaluator) to guide the feature set embedding generation towards performance maximization. The reward guidance is interpreted as aligning diffusion trajectories with reward optimization paths to combine global distribution with local refinement. To overcome the limitations of autoregressive decoding, we propose a hierarchical semi-autoregressive decoding method. We divide a feature token sequence into multiple chunks (e.g., "$f_1 \ f_2 \ *$, $f_3 \ \log, \ f_4 \ f_5 \ +$" has 3 chunks), each of which is one transformed feature (e.g., "$f_1 \ f_2 \ *$" is one feature). We adopt a hierarchical strategy to firstly predict how many feature chunks to generate and then autoregress the token sequence of each chunk. This can enable parallel decoding to improve efficiency of decoding a feature token sequence for transformation. Collectively, these insights allow us to frame FT not as a search problem within a discrete or continuous space, but as a guided generative task. Figure 1 visually encapsulates this paradigm shift. It contrasts our diffusion-based generation approach with prior discrete and continuous search methodologies, highlighting how leveraging the global sampling

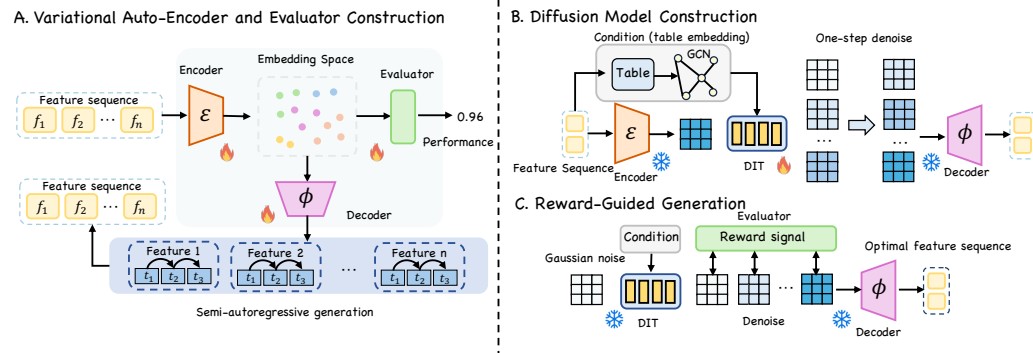

Figure 2: Framework overview. The framework consists of three key components: 1) a VAE that encodes feature sequences into latent embeddings via a semi-autoregressive decoder; 2) a LDM trained to model the distribution of effective embeddings conditioned on tabular semantics; and 3) a reward-guided sampling process that leverages gradients from a performance evaluator to steer the generation of high-quality feature embeddings, which are then decoded into final feature sets.

capabilities of diffusion and reward guidance can overcome the inherent limitations of large search spaces and local optima that challenged previous techniques.

**Summary of Proposed Approach.** Inspired by these insights, we propose a Reward-Guided Diffusion Framework for Semi-Autoregressive Generative FT (**DIFFT**). Our method includes a Variational Auto-Encoder (VAE) that learns compact embeddings of feature token sequences, an evaluator network that guides the sampling process during inference in reverse diffusion, and a Latent Diffusion Model (LDM) that generates task-optimal feature set embeddings within the learned latent space. In particular, given tasking data, we explore and collect the performance of different transformed feature sets (represented by feature token sequences) as training data. The VAE encoder is trained to encode these feature token sequences into embedding vectors, while the VAE decoder predicts the number of features in the output feature token sequence, then autoregresses each transformed feature in a step-by-step manner. After training the VAE, the feature token sequence embedding vectors of training data are exploited to learn the LDM as a denoising generative model in the embedding space. After training the LDM, we extract gradient information from the evaluator that estimates the performance of feature set sequence embeddings to guide the reverse denoising diffusion to generate (a.k.a, sample) task-optimal feature token sequence embeddings. Finally, we use the VAE decoder to decode the task-optimal embedding into a transformed feature set to augment input data.

**Our Contributions.** 1) We tackle FT as a GenAI task and develop an integrated reward-guided diffusion and hierarchical decoding paradigm. 2) We convert task-optimal FT search as evaluator-steered reverse diffusion, in which diffusion models learn feature set embedding distribution, then conduct generative sampling of the desirable embedding. 3) We design a semi-autoregressive decoding strategy (i.e., predicting feature number then generating token sequence of each feature) to parallelize decoding and advance efficiency.

## 2 Diffusion Feature Transformation

### 2.1 Framework Overview

Figure 2 shows that our method has three core components: (1) a Variational Auto-Encoder for feature knowledge learning; (2) a Latent Diffusion Model for embedding generation; and (3) a reward-guided sampling mechanism for performance-aware feature transformation. In step 1, we construct a VAE with an encoder that maps discrete feature sequences into a compact latent space, and a semi-autoregressive decoder that reconstructs each feature sequence by first predicting the number of features and then generating each feature autoregressively. Additionally, we train a performance evaluator to predict the utility of latent embeddings for downstream tasks. In step 2, we use the VAE encoder to obtain latent representations of training feature sets and train an LDM as a denoising generative model to capture the distribution of effective embeddings. In step 3, during inference, we incorporate the evaluator's gradients into the denoising process, guiding the diffusion model to generate embeddings that are optimized for downstream performance. The generated embeddings are

then decoded via the semi-autoregressive decoder to produce high-quality feature sets. We provide more information on important concepts in Appendix A.

## 2.2 Embedding: Learning the Embedding Space of Feature Sets on Tasking Data

We construct a representation learning model (i.e., a variational probabilistic autoencoder structure) to encode discrete feature token sequences into embedding vectors in a compact and informative embedding space, and a feature utility evaluator to predict the performance of a feature set given the corresponding feature token sequence embedding. Figure 2(A) shows our model includes an encoder, an evaluator, and a decoder.

**Step 1: Transformer-based Seq2Vec Encoder.** Given a discrete feature token sequence $\mathcal{F} = \{f_1, f_2, \ldots, f_T\}$, the encoder $E_\phi(\cdot)$ encodes the feature sequence into a latent vector $\mathbf{z}$, which is sampled from a latent distribution $q_\phi(\mathbf{z}|\mathcal{F}) = \mathcal{N}(\mu, \sigma^2)$. To reshape the distribution of the latent space into Gaussian, we impose a KL divergence to measure and minimize the distance between the embedding space posterior distribution and a standard Gaussian prior:

$$\mathcal{L}_{\text{KL}} = \text{KL}(q_\phi(\mathbf{z}|\mathcal{F})\|p(\mathbf{z})). \tag{1}$$

**Step 2: Hierarchical Decoder with Position Embedding.** The decoder is to generate a feature sequence from a given latent vector $\mathbf{z}$. When the decoder's input is the decoder's output embedding, the task is reduced to reconstruct the original feature set. Unlike natural languages, a feature set, although tokenized as a feature token sequence, has no current-past feature dependencies. For instance, in the feature sequence $f_1 + f_2, f_2/f_3$, the second feature token segmentation $f_2/f_3$ is not dependent on the first feature token segmentation $f_1 + f_2$. Only the token sequence within a single feature segmentation (e.g., $f_1 + f_2$) exhibits sequential dependencies. We name a tokenized feature set as a compositional sequence.

To model the compositional sequence structure, we propose a hierarchical decoder that adopts a two-level (inter-feature and intra-feature) generation strategy. Specifically, at the inter-feature level, the decoder predicts how many features we should generate as a feature set, which is modeled as a categorical distribution using the embedding of the feature set. Formally, let $T$ be the number of features, $p_\theta$ denote the decoder's conditional probability distribution parameterized by $\theta$, the feature size prediction is trained by minimizing a standard cross-entropy loss:

$$\mathcal{L}_{cot} = -\log p_\theta(T|\mathbf{z}). \tag{2}$$

After predicting the size of the feature set to decode, the decoder independently generates each feature in parallel. At the intra-feature level, the decoder autoregressively generates each feature token segmentation starting with a special token of "BOS". Formally, let $f_t^{(k)}$ be the $k$-th token of the $t$-th feature, $L_t$ is the total token number of $f_t$. The feature token sequence reconstruction is learned by optimizing a token-level cross-entropy loss over all the predicted features:

$$\mathcal{L}_{rec} = -\sum_{t=1}^{T} \sum_{k=1}^{L_t} \log p_\theta(f_t^{(k)}|f_t^{(<k)}, \mathbf{z}). \tag{3}$$

**Step 3: Incorporating The Evaluator.** To incorporate performance guidance, we construct a lightweight regression evaluator to predict the downstream task performance of a feature set, based on the feature set's embedding. Formally, $R_\psi(\cdot)$ is a regression function, $y$ is the real downstream task performance of a feature set. The evaluator is trained by minimizing a mean squared error loss:

$$\mathcal{L}_{eva} = \|R_\psi(\mathbf{z}) - y\|^2, \tag{4}$$

**Solving the Optimization Problem.** The encoder-decoder-evaluator structure is trained end-to-end by jointly optimizing the losses of Gaussian regularization, feature set size prediction, feature set reconstruction, and feature set performance evaluation:

$$\mathcal{L} = \mathcal{L}_{rec} + \alpha\mathcal{L}_{cot} + \beta\mathcal{L}_{eva} + \gamma\mathcal{L}_{KL}, \tag{5}$$

where $\alpha$, $\beta$, and $\gamma$ are the weights of corresponding loss terms.

**Finally**, after the encoder, decoder, and evaluator are trained, the encoder can create a continuous embedding space to map various transformed feature sets on a specific dataset into embedding vectors,

thus laying a foundation for the diffusion model to learn and generalize the distributions from these embedding vectors; the evaluator can provide performance-optimal directions to enable diffusion models to sample (a.k.a., generate) the best transformed feature set embedding from the distribution via maximized likelihood estimation; the decoder, if well trained, can decode the best transformed feature set embedding into a feature token sequence.

## 2.3 Distribution: Diffusion Model of Probabilistic Feature Embedding Space

After completing the embedding step, we utilize the learned encoder to project various feature token sequences (i.e., potential feature transformations) into embedding vectors. The embedding vectors of potential feature transformations can be seen as samples that jointly form a distribution about how feature performances are probabilistically spread across various feature set embeddings. This provides a new probabilistic angle: if we can describe the distribution of feature set embeddings over performance, we can identify the best feature transformation plan for a dataset by generating the mean feature set embedding with the highest probability from the distribution. The Latent Diffusion Model (LDM) exhibits the ability to learn distributions from noisy and limited data in reverse diffusion and generate performant samples from the distribution. This ability provides great potential for us to derive the distribution of feature set embeddings and identify the best feature set transformation. We next introduce the feature set embedding diffusion model.

**Leveraging Feature Sets as Attributed Graphs and Graph Embedding for Diffusion Conditioning.** To steer diffusion generation, we condition the diffusion process on an external representation that captures distributional semantics of the original raw dataset to transform. Specifically, we convert a feature token sequence into a tabular modality, by applying the feature token sequence to the raw data matrix, resulting in a transformed table. Based on the transformed table, we construct a feature-feature attributed graph, where nodes are features (columns) and edge weights are feature-feature similarities to represent tabular feature space topology, interaction, and semantics. We then exploit a Graph Convolutional Network (GCN) to learn an embedding of this graph. In this way, a feature token sequence, corresponding to its transformed table and feature-feature interaction graph, is associated with a graph embedding vector. After that, we leverage this graph embedding as a conditioning signal during both training and sampling. Such conditioning strategy allows the diffusion model to generate embedding vectors aligned with the structure, topology, and semantics of the targeted tasking dataset to transform, enabling coherent and meaningful feature generation. Appendix B provides details about the diffusion condition acquisition.

**Learning Feature Transformation Diffusion Model.** Because we regard a transformed feature set as a feature token sequence, not an image, we choose a Transformer-based denoiser [33], instead of the conventional UNet architecture. Our diffusion model includes a Transformer-encoder of $L$ blocks that apply sequentially to the current latent state $\mathbf{z}_t$. For the $\ell$-th block, we compute

$$\mathbf{h}^{(\ell,1)} = \text{SA}\big(\text{AdaLN}_t(\mathbf{h}^{(\ell-1)})\big), \tag{6}$$

$$\mathbf{h}^{(\ell,2)} = \text{CA}\big(\text{AdaLN}_t(\mathbf{h}^{(\ell,1)}), \mathbf{c}\big), \tag{7}$$

$$\mathbf{h}^{(\ell)} = \text{FFN}\big(\mathbf{h}^{(\ell,2)}\big), \tag{8}$$

where **SA** is self-attention over latent tokens, **CA** cross-attends to the condition embedding $\mathbf{c}$, and **FFN** is a two-layer feed-forward network. The timestep $t$ enters through the AdaLayerNorm modulation [21], and the condition $\mathbf{c} = \text{MLP}_{\text{cond}}(\mathbf{g})$ is obtained by passing the tabular embedding $\mathbf{g}$ through slight MLPs. The network predicts the noise $\boldsymbol{\epsilon}_\theta(\mathbf{z}_t, t, \mathbf{c})$ and is trained with the standard latent diffusion loss $\mathcal{L}_{\text{LDM}} = \mathbb{E}\big[\|\boldsymbol{\epsilon} - \boldsymbol{\epsilon}_\theta(\mathbf{z}_t, t, \mathbf{c})\|_2^2\big]$. Appendix D.4 details the training of the diffusion model.

## 2.4 Optimization as Generative Sampling: Reward-guided Optimal Feature Set Embedding Generation

Given a target reward $a$ that indicates a high utility of features to generate, we follow the gradient-steered sampler of RCGDM [32] to consider the latent z as Gaussian and develop our reverse process. At each denoising step, we first evaluate $\nabla_{\mathbf{z}} \log p_t(y = a \mid \mathbf{z}) \approx -\frac{1}{\sigma^2} \nabla_{\mathbf{z}} \big[\frac{1}{2}(R_\psi(\mathbf{z}) - a)^2\big]$, where $R_\psi$ is the pretrained evaluator and $\sigma^2$ is a hyper-parameter. The DDIM update [22] then becomes

$$\mathbf{z}_{t-1} = \sqrt{\alpha_{t-1}}\Big(\mathbf{z}_t - \frac{1 - \alpha_t}{\sqrt{1 - \bar{\alpha}_t}}\big[\boldsymbol{\epsilon}_\theta(\mathbf{z}_t, t, \mathbf{c}) - \lambda\nabla_{\mathbf{z}}\big(\tfrac{1}{2}(R_\psi(\mathbf{z}_t) - a)^2\big)\big]\Big) + \sqrt{1 - \alpha_{t-1}}\,\boldsymbol{\eta}_t, \tag{9}$$

with $\lambda = 1/\sigma^2$ controlling the strength of reward guidance and $\boldsymbol{\eta}_t \sim \mathcal{N}(\mathbf{0}, \mathbf{I})$ the optional stochasticity. Note that we do not apply classifier-free guidance [10] in our model, since the reward guidance we employ shares the same fundamental principle as classifier-based guidance, which is approximating the conditional score $\nabla \log p_t(x_t \mid y)$ [5]. Upon completion, the final latent $\hat{\mathbf{z}}_0$ is decoded by the semi-autoregressive decoder to yield a feature set that is syntactically valid and empirically high-reward.

### 2.5 Hierarchical Decoding: Semi-autoregressive Feature Set Reconstruction from Optimal Embedding

After the reverse process samples or generates the optimal high-reward feature set embedding from the embedding distribution, we leverage the learned decoder to decode the optimal embedding into a feature token sequence, in order to obtain a new transformation. The hierarchical decoding includes two steps: 1) predict feature size; 2) autoregress the token sequence of each feature chunk. Formally, the decoder first predicts the number of features $T$:

$$T \sim p_\theta(T \mid \hat{\mathbf{z}}_0).$$

For each feature chunk $f_t$, the decoder generates its token sequence in an autoregressive manner:

$$f_t^{(k)} \sim p_\theta(f_t^{(k)} \mid f_t^{(<k)}, \hat{\mathbf{z}}_0), \quad k = 1, \dots, L_t.$$

This semi-autoregressive decoding strategy enables parallel generation across feature chunks while maintaining autoregressive modeling within each feature, balancing efficiency and generation quality.

## 3 Experiments

We conduct extensive experiments on public datasets to evaluate the effectiveness, efficiency, and robustness of our method. The experiments are organized to answer the following research questions: **RQ1:** Does the proposed method perform better than the baselines? **RQ2:** Do reward-guided diffusion and semi-autoregressive generation contribute to the overall effectiveness and efficiency of the framework? **RQ3:** Is the proposed method efficient compared with baselines? **RQ4:** Is the proposed method robust when collaborating with different downstream ML models? **RQ5:** How does guidance strength in generation affect performance?

### 3.1 Experimental Setup

We collect 14 public datasets from LibSVM, UCIrvine, and OpenML to conduct experiments. The datasets are classified into two predictive tasks: classification and regression. We employ random forest as the default downstream model and use F-1 score and 1- relative absolute error (RAE) to measure the performance of classification task and regression task respectively. We compare the proposed method with 10 widely-used algorithms: RDG, PCA [18], LDA [3], ERG, AFAT [11], NFS [4], TTG [15], GRFG [26], MOAT [27], ELLM [6]. More details about datasets and baselines can be found in Appendix D.

To comprehensively evaluate the effectiveness of each component, we introduce four ablation variants of DIFFT: 1) *AR* employs a fully autoregressive generation strategy in the VAE decoder, replacing our semi-autoregressive design; 2) *NAR* adopts a fully non-autoregressive decoder to assess the importance of intra-feature dependency modeling; 3) *NoR* removes reward guidance during diffusion to examine the role of performance-aware optimization; 4) *CS* eliminates the diffusion model and instead applies a conventional continuous search method in the latent space to validate the effectiveness of the diffusion model.

### 3.2 Overall Comparison (RQ1)

Table 1 reports the performance across 14 benchmark datasets. DIFFT consistently achieves the best results on all datasets, demonstrating strong effectiveness and robustness across a wide range of tasks. For example, on Messidor Features, DIFFT achieves an accuracy of 0.757, substantially higher than the second-best method ELLM with 0.699. On Openml_618, DIFFT reaches 0.632, outperforming MOAT with 0.477 and other baselines by a significant margin. This performance

Table 1: Overall comparison. We conduct experiments on 14 datasets to explore the performance of the proposed method. The best results for each dataset are highlighted in bold, and the second-best results are underlined.

| Dataset | RDG | PCA | LDA | ERG | AFAT | NFS | TTG | GRFG | MOAT | ELLM | **DIFFT** |
|---|---|---|---|---|---|---|---|---|---|---|---|
| SpectF | 0.760 | 0.709 | 0.665 | 0.792 | 0.792 | 0.760 | 0.776 | 0.796 | 0.848 | 0.837 | **0.877** |
| SVMGuide3 | 0.789 | 0.676 | 0.635 | 0.764 | 0.795 | 0.785 | 0.766 | 0.804 | 0.821 | 0.826 | **0.866** |
| German Credit | 0.657 | 0.679 | 0.597 | 0.711 | 0.639 | 0.677 | 0.680 | 0.722 | 0.738 | 0.741 | **0.747** |
| Messidor Features | 0.686 | 0.672 | 0.463 | 0.602 | 0.634 | 0.649 | 0.626 | 0.694 | 0.644 | 0.699 | **0.757** |
| SpamBase | 0.921 | 0.816 | 0.885 | 0.923 | 0.920 | 0.925 | 0.921 | 0.928 | 0.930 | 0.928 | **0.931** |
| Ap-omentum-ovary | 0.849 | 0.736 | 0.696 | 0.830 | 0.845 | 0.845 | 0.830 | 0.830 | 0.849 | 0.849 | **0.885** |
| Ionosphere | 0.898 | 0.928 | 0.743 | 0.914 | 0.899 | 0.899 | 0.927 | 0.928 | 0.928 | 0.942 | **0.971** |
| Openml_586 | 0.532 | 0.205 | 0.061 | 0.542 | 0.549 | 0.553 | 0.551 | 0.550 | 0.607 | 0.611 | **0.647** |
| Openml_589 | 0.509 | 0.222 | 0.033 | 0.502 | 0.507 | 0.503 | 0.503 | 0.590 | 0.510 | 0.595 | **0.615** |
| Openml_607 | 0.521 | 0.106 | -0.040 | 0.518 | 0.513 | 0.518 | 0.518 | 0.524 | 0.521 | 0.550 | **0.561** |
| Openml_616 | 0.120 | -0.083 | -0.174 | 0.193 | 0.163 | 0.166 | 0.161 | 0.166 | 0.162 | 0.329 | **0.335** |
| Openml_618 | 0.425 | 0.141 | 0.030 | 0.470 | 0.473 | 0.473 | 0.472 | 0.472 | 0.477 | 0.476 | **0.632** |
| Openml_620 | 0.502 | 0.138 | -0.045 | 0.501 | 0.520 | 0.507 | 0.515 | 0.528 | 0.510 | 0.550 | **0.592** |
| Openml_637 | 0.136 | -0.052 | -0.141 | 0.152 | 0.162 | 0.146 | 0.145 | 0.153 | 0.161 | 0.254 | **0.291** |

is primarily driven by two key components: the latent diffusion model, which enables global exploration of the embedding space to discover high-quality feature representations, and the semi-autoregressive decoder, which efficiently reconstructs complex feature structures with high fidelity. Compared to MOAT, the representative continuous search-based method, DIFFT offers significantly better stability and generalization. Although MOAT occasionally produces competitive results, its performance fluctuates considerably across datasets and drops sharply on more challenging tasks such as Openml_616, where it reaches only 0.162 compared to 0.335 achieved by DIFFT. In contrast, DIFFT performs consistently well, benefiting from its generative formulation and reward-guided sampling strategy. By avoiding local search and leveraging global, performance-aware generation, DIFFT delivers more reliable and superior feature transformations across diverse tasks.

## 3.3 Ablation Study (RQ2)

To investigate the contribution of each component in DIFFT, we design variant models: 1) *NAR* and *AR*, which examine the impact of the semi-autoregressive decoder by replacing it with fully non-autoregressive and fully autoregressive alternatives, respectively; 2) *NoR*, which removes reward guidance during the diffusion process to evaluate its effect on performance optimization; and 3) *CS*, which replaces the diffusion model with a traditional continuous search approach to assess the necessity of diffusion-based generation. We conduct ablation study on two datasets to validate the effectiveness of the components. More details can be found in Appendix D.5.

Table 2: Performance of variant models and DIFFT on 2 datasets.

| Dataset | Variants | | | | DIFFT |
|---|---|---|---|---|---|
| | NAR | AR | NoR | CS | |
| SVMGuide3 | 0.840 | 0.856 | 0.831 | 0.830 | **0.866** |
| Openml_586 | 0.641 | 0.638 | 0.623 | 0.610 | **0.647** |

**Reward-Guided Diffusion.** As shown in Table 2, both *NoR* and *CS* suffer from significant performance drops compared to the full model. For example, on the SVMGuide3 dataset, removing reward signals (*NoR*) reduces the accuracy from 0.866 to 0.831, while replacing diffusion with continuous search (*CS*) causes an even greater decline to 0.830. This highlights the critical role of reward guidance and the diffusion mechanism in driving feature generation toward high-performing regions of the latent space. Interestingly, even without reward guidance, *NoR* still outperforms the continuous search-based variant *CS* on the two datasets. This suggests that in the absence of a carefully constructed latent space and an informative initialization, continuous search methods often fail to explore effectively and are prone to suboptimal solutions. In contrast, by modeling global structure and semantics, our diffusion model provides a more robust foundation for feature embedding generation.

**Semi-Autoregressive Generation.** The results in Table 2 also demonstrate the effectiveness of the proposed semi-autoregressive (*SAR*) decoder. Compared with both *AR* and *NAR* variants, *SAR* achieves superior performance across datasets. For example, on SVMGuide3, the *SAR* decoder achieves an accuracy of 0.866, surpassing both the fully autoregressive model with 0.856 and the non-autoregressive variant with 0.840. This improvement demonstrates that *SAR* effectively balances dependency modeling and generation efficiency. By generating each feature independently while preserving intra-feature autoregression, *SAR* captures fine-grained dependencies within individual features, while avoiding redundant coupling across features that are semantically unrelated. This selective dependency modeling enables the decoder to allocate attention more efficiently, resulting in more stable and interpretable feature generation.

To further analyze efficiency, Figure 3 explores the decoding time of *SAR* decoders under different configurations to generate 2000 tokens. We vary the number of features and the length of each feature such that the total token count remains constant (i.e., number × length). The results show that *SAR* consistently achieves significantly lower decoding time than *AR*, regardless of the configuration. Although decoding time slightly increases with longer features, our method maintains high efficiency by parallelizing feature generation.

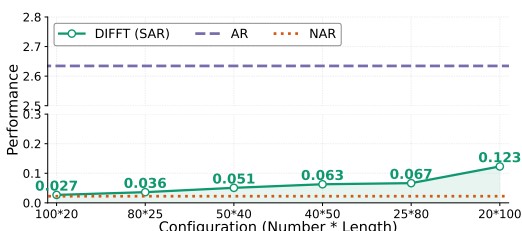

Figure 3: Time analysis of generating 2000 tokens using different methods.

## 3.4 Time Complexity Analysis (RQ3)

Table 3: Comparison of per-epoch training time and per-inference pass time (in seconds) between DIFFT and MOAT across different datasets. DIFFT (D) and DIFFT (V) denote the per-epoch training time of the diffusion model and VAE, respectively, while inference time refers to the duration of one complete inference pass.

| Dataset | Feature # | Training Time (s) | | | | Inference Time (s) | |
|---|---|---|---|---|---|---|---|
| | | DIFFT (D) | DIFFT (V) | DIFFT | MOAT | DIFFT | MOAT |
| Messidor Features | 19 | 2.65 | 4.36 | 7.01 | 83.9 | 0.16 | 0.53 |
| Openml_589 | 25 | 2.85 | 5.02 | 7.87 | 110.9 | 0.21 | 0.68 |
| Ionosphere | 34 | 3.33 | 5.35 | 8.68 | 170.1 | 0.23 | 1.03 |
| Openml_618 | 50 | 4.74 | 7.45 | 12.19 | 200.9 | 0.25 | 1.18 |
| Ap_omentum_ovary | 10936 | 7.78 | 11.86 | 19.64 | 524.4 | 0.43 | 2.33 |

We compare the training and inference efficiency of DIFFT and MOAT to evaluate the computational characteristics of the two generative feature transformation methods. Table 3 reports the per-epoch training time and per-instance inference time for both methods. For DIFFT, we separately record the training time of the VAE (denoted as DIFFT (V)) and the diffusion model (denoted as DIFFT (D)). DIFFT requires significantly less time to train per epoch compared to MOAT. For example, on the large-scale Ap_omentum_ovary dataset with over 10,000 features, DIFFT completes one training epoch in 19.64 seconds, while MOAT takes more than 500 seconds. This efficiency gain is largely attributed to architectural differences: DIFFT adopts a Transformer-based backbone, which supports efficient parallel computation and faster convergence, whereas MOAT relies on an LSTM-based model with sequential processing and limited scalability. Inference speed further highlights the efficiency of our approach. DIFFT generates a feature set in no more than 0.43 seconds on all datasets, whereas MOAT takes up to 2.33 seconds in some cases. This leads to a speedup of more than 5 times, mainly due to the semi-autoregressive decoding strategy that allows parallel feature-level generation while maintaining high generation quality. These results demonstrate that DIFFT offers not only strong performance but also substantial advantages in training and inference efficiency.

## 3.5 Robustness Check (RQ4)

To evaluate the robustness of DIFFT across different learning scenarios, we apply five representative downstream machine learning models on the SpectF dataset, including Decision Tree (DT), k-Nearest Neighbors (KNN), Logistic Regression (LR), Support Vector Classifier (SVC), and Random Forest (RF). Table 4 reports the performance of DIFFT compared with the top three baselines. We observe that DIFFT consistently achieves the best performance across all downstream models, demonstrating strong robustness to variations in model architectures and learning biases. In contrast, ELLM performs reasonably well on some models but fails to achieve competitive results on RF, indicating limited generalization. Similarly, MOAT performs poorly when paired with SVC, suggesting sensitivity to the characteristics of the downstream learner. These findings highlight the robustness of DIFFT in optimizing feature transformation across diverse tasks. DIFFT generates task-specific features that align with the objective of the target model, enabling consistent improvements in predictive performance across a wide range of learning scenarios.

Table 4: Robustness check. We employ different downstream ML model to investiagte the robustness of our method.

| Method | DT | KNN | LR | SVC | RF |
|---|---|---|---|---|---|
| GRFG | 0.752 | 0.764 | 0.816 | 0.813 | 0.796 |
| MOAT | 0.873 | 0.843 | 0.835 | 0.836 | 0.848 |
| ELLM | 0.865 | 0.842 | 0.846 | 0.841 | 0.837 |
| **DIFFT** | **0.894** | **0.861** | **0.864** | **0.877** | **0.877** |

### 3.6 Guidance Strength Analysis (RQ5)

Table 5: Performance under different guidance strength $\lambda$ settings.

| Dataset | 0 | 10 | 25 | 50 | 100 | 200 | 400 |
|---|---|---|---|---|---|---|---|
| SVMGuide3 | 0.831 | 0.857 | 0.867 | 0.874 | 0.871 | 0.860 | 0.860 |
| Openml_616 | 0.237 | 0.262 | 0.232 | 0.278 | 0.324 | 0.302 | 0.170 |

To investigate the effect of the reward-guided strength $\lambda$ on the generation performance, we conduct experiments on two datasets, with $\lambda$ ranging from 0 to 400. The results are summarized in Table 5. For SVMGuide3, we observe a clear improvement as $\lambda$ increases from 0 to 50, where the performance peaks at 0.874. This demonstrates that moderate guidance helps the diffusion process align more effectively with the reward objective. However, when $\lambda$ becomes too large (e.g., $\geq$ 100), the performance slightly decreases, suggesting that excessive guidance may overconstrain the generation, leading to reduced sample diversity and potential overfitting to the reward signal. For Openml_616, the trend is less monotonic but shows a similar pattern overall. The performance improves gradually and achieves its highest value (0.324) at $\lambda = 100$, followed by a decline when $\lambda$ increases further. The larger variability across $\lambda$ values implies that this dataset may be more sensitive to the balance between reward alignment and generative flexibility. Overall, these results indicate that moderate guidance strength (around $\lambda$ = 50–100) yields the best trade-off between reward optimization and sample diversity. Too weak guidance fails to leverage the reward signal effectively, while too strong guidance can hinder the generative capacity of the diffusion model.

## 4 Related Work

### 4.1 Feature Transformation

Feature transformation aims to identify a new feature space by transforming the original features using mathematical operations. The existing methods are mainly two-fold: 1) discrete space search methods. These methods aim to search for the optimal feature set in the discrete feature combination space. DFS [13] transforms all original features and selects useful ones. Genetic programming is applied for feature transformation search [24]. GRFG [26] develops three agents to generate features and leverages reinforcement learning to improve the search strategy. ELLM [6] employs few-shot prompting LLM to generate new feature sets. 2) continuous space search methods. MOAT [27] embeds feature sets into continuous space and utilizes the gradient-ascent method to explore the embedding space. NEAT [30] extends the continuous space feature transformation search to unsupervised conditions.

## 4.2 Diffusion Model

Diffusion models have emerged as a dominant paradigm in generative modeling, offering superior sample quality and training stability compared to traditional approaches such as GANs and VAEs. Their fundamental design involves a forward noising process and a learned reverse process that enables high-fidelity generation across modalities. Conditional variants of diffusion models, which incorporate task-specific signals such as class labels, text prompts, or reward functions, have further extended their applicability to controlled generation tasks in vision, language, and reinforcement learning. Recent applications span a diverse range of domains, including text-to-image synthesis [2], trajectory planning in reinforcement learning [12, 1], and protein or molecule design in life sciences [8]. Beyond applications, a growing line of theoretical research has addressed key questions around score function learning, sample complexity, and optimization guarantees, especially in high-dimensional and structured settings [19]. Recent works also reinterpret conditional diffusion models as flexible samplers for solving black-box optimization problems, where solution quality is guided by reward-driven conditioning [17]. Despite the growing empirical success, theoretical analysis remains an open frontier, particularly regarding guidance strength and generalization under distribution shifts.

## 4.3 Semi-Autoregressive Model

To achieve a better trade-off between decoding speed and output quality, semi-autoregressive generation (SAR) has been proposed as a hybrid decoding paradigm that blends the strengths of both autoregressive (AR) and non-autoregressive (NAR) models. Early work by Wang et al. introduced the semi-autoregressive Transformer, which generates tokens in chunks—producing multiple tokens in parallel within each group, while maintaining an autoregressive dependency across groups [25]. This design allows for partial parallelization while preserving sufficient contextual dependency to mitigate issues such as repetition or omission in NAR outputs. Building upon this, the Insertion Transformer supports flexible generation orders by allowing insertions at arbitrary positions, offering a more dynamic decoding process [23]. Other efforts, such as the DisCo Transformer [14], further relax strict left-to-right generation by enabling token predictions conditioned on arbitrary subsets of other tokens. Additionally, RecoverSAT [20] introduces a segment-wise decoding strategy where each segment is generated non-autoregressively, while the tokens within segments follow an autoregressive pattern. These semi-autoregressive frameworks consistently demonstrate a superior balance between latency and accuracy compared to fully NAR models, making them attractive choices in time-sensitive applications.

# 5   Conclusion Remarks

In this paper, we propose DIFFT, a novel framework that formulates feature transformation as a reward-guided generative process. Departing from traditional search-based paradigms, DIFFT integrates a latent diffusion model with a semi-autoregressive decoder to generate high-quality, task-specific feature sets in a performance-aware manner. Our hierarchical decoding strategy enables efficient and expressive feature reconstruction, while the reward-guided sampling process aligns the generative trajectory with downstream objectives. Extensive experiments on a diverse collection of tabular datasets demonstrate that DIFFT consistently outperforms existing feature transformation and selection methods in terms of predictive performance, robustness across downstream models, and computational efficiency. Our ablation studies further validate the contributions of each module, including the semi-autoregressive decoding and reward-driven diffusion. These results highlight the effectiveness and adaptability of DIFFT as a scalable solution for feature engineering in modern machine learning pipelines.

# Acknowledgments

Dr. Yanjie Fu is supported by the National Science Foundation (NSF) via the grant numbers: 2426340, 2416727, 2421865, and 2421803.

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

# A  Preliminaries

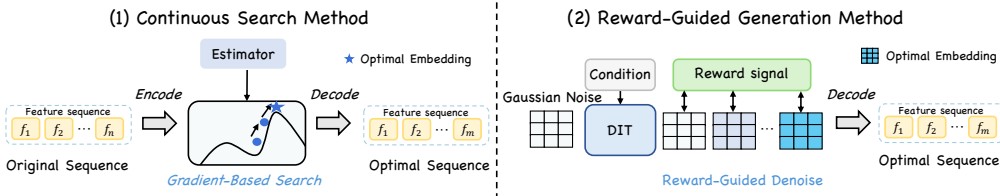

Figure 4: Comparison of continuous search method and optimized generation method.

**Feature Transformation.** Given a raw feature set $\mathcal{F}_{\text{raw}} = [f_1, f_2, \ldots, f_n]$ and an operator set $\mathcal{O}$ (e.g., $\log, +, *$), feature transformation aims to apply mathematical operations to original features and construct a better feature set, for instance, $[f_1 * f_2, \log f_3, f_4 + f_5]$. These transformed features reveal informative interactions and nonlinear relationships that improve downstream task performance.

**Feature Sets as Language Token Sequences.** A feature set can be described in a language token modality. For example, given a transformed feature set $[f_1 * f_2, \log f_3, f_4 + f_5]$, we use a token sequence "$f_1 * f_2, \log f_3, f_4 + f_5$" to represent it in a compact and generative-friendly form. To reduce sequence length and ambiguity, we adopt the postfix notion: "$f_1 \ f_2 \ *, f_3 \ \log, f_4 \ f_5 \ +$".

**Generative Feature Transformation and the Continuous Search Paradigm.** Classic feature engineering relies on manual selection or discrete search over massive possible combinations. With the success of generative AI and everything as tokens, researchers regard a feature set as a token sequence, and formulates feature transformation as a sequential generation task: given X and y, generate a new feature token sequence (i.e., a new transformed feature set) that reconfiguring the data that the model sees to align model behavior with prediction targets. The major methodological paradigm under the generative formulation is continuous search that consists of three steps: 1) embed feature token sequence into a continuous embedding space; 2) apply gradient ascent to discover better embedding of feature transformations; 3) decode identified embedding to a new table. Building on this idea, we propose a novel reward-guided generation framework that goes beyond searching within a static embedding space: instead, it directly generates and refines feature representations through a performance-aware generative process. This enables dynamic, goal-oriented exploration of the transformation space, as summarized in Figure 4.

**Latent Diffusion Model.** Diffusion models [22, 21] synthesize data by learning to reverse a gradual noising process that turns a clean sample into Gaussian noise. Let $z_0$ be a latent drawn from the data distribution. The forward (noising) process adds variance-preserving Gaussian noise for $T$ steps

$$q(z_t \mid z_{t-1}) = \mathcal{N}\big(z_t; \sqrt{1 - \beta_t}\, z_{t-1}, \beta_t \mathbf{I}\big), \qquad t = 1, \ldots, T,$$

with a predefined schedule $\{\beta_t\}_{t=1}^{T}$. A neural network $\epsilon_\theta$ is trained to approximate the reverse (denoising) transitions by predicting the added noise:

$$\mathcal{L}_{\text{DM}} = \mathbb{E}_{t, z_0, \epsilon}\Big[\big\|\epsilon - \epsilon_\theta(z_t, t)\big\|_2^2\Big], \qquad z_t = \sqrt{\bar{\alpha}_t}\, z_0 + \sqrt{1 - \bar{\alpha}_t}\, \epsilon, \ \bar{\alpha}_t = \prod_{s=1}^{t}(1 - \beta_s).$$

In latent diffusion, data are first compressed with an auto-encoder $(\text{Enc}, \text{Dec})$ into a perceptually aligned latent space, and the diffusion process operates entirely on these low-dimensional latents. Compared with pixel-/token-space diffusion, LDMs (i) drastically reduce memory and computation, (ii) facilitate conditioning on vector-space signals, and (iii) retain high-fidelity details because decoding starts from semantically rich latents instead of raw noise. During inference, one samples $z_T \sim \mathcal{N}(\mathbf{0}, \mathbf{I})$ and applies the learned reverse transitions to obtain a clean latent $\hat{z}_0$, which is mapped back to the data domain by $\text{Dec}(\hat{z}_0)$. Importantly, each denoising step is differentiable, so external gradients—e.g., from a reward or performance predictor—can be injected to *steer* the trajectory toward regions that maximize downstream objectives.

# B  Condition Acquisition

To provide a compact and informative conditioning signal for the diffusion model, we extract a table-level embedding that summarizes the structure of the transformed tabular data. The feature

sequence is interpreted as a table, where each column corresponds to a transformed feature. This table structure captures both individual feature characteristics and inter-feature statistical dependencies.

We adopt a GCN-based encoder $\phi$ to generate the table embedding. Specifically, each feature is regarded as a node in a fully connected feature correlation graph with self-loops. The adjacency matrix $\hat{A}$ encodes the pairwise correlations, and the node input features form the matrix $H^{(0)}$. We apply a simplified Graph Convolutional Network [16] using symmetric normalization:

$$f(H^{(l)}, A) = \sigma\left(\hat{D}^{-\frac{1}{2}} \hat{A} \hat{D}^{-\frac{1}{2}} H^{(l)} W^{(l)}\right) \tag{10}$$

where $\hat{A}$ is the adjacency matrix with self-loops, $\hat{D}$ is the degree matrix, $W^{(l)}$ is the trainable weight matrix at layer $l$, and $\sigma$ is a non-linear activation function. After GCN, we obtain contextualized embeddings for all features. These are aggregated via average pooling to obtain a table-level embedding:

$$e_t = \frac{1}{n} \sum_{i=1}^{n} h_i \tag{11}$$

where $h_i$ denotes the final embedding of the $i$-th feature. The resulting $e_t$ is then used as the condition input to the diffusion model during both training and generation, enabling it to guide the feature generation process based on the global structure of the table.

## C  Training Data Collection

Generative feature transformation requires a diverse and high-quality training dataset to construct a meaningful embedding space. However, traditional manual data collection is often inefficient and lacks scalability. To address this challenge, we design a reinforcement learning (RL)-based data collector that automatically explores and generates effective feature transformations, inspired by recent advances in automated feature engineering [6, 26].

The proposed RL-based data collector simulates feature transformation trajectories and consists of the following components:

- **1) Agents:** A multi-agent system is employed, comprising:
    - *Head Feature Agent $\alpha_h$*: selects the first feature.
    - *Operation Agent $\alpha_o$*: selects an operator from the operator set $\mathcal{O}$.
    - *Tail Feature Agent $\alpha_t$*: selects the second feature.

- **2) Actions:** At each time step $t$, the three agents collaboratively generate a new feature as:

$$f_t = \alpha_h(t) \oplus \alpha_o(t) \oplus \alpha_t(t) \tag{12}$$

   where $\oplus$ denotes the operator applied to the selected feature pair.

- **3) State Representation:** The state $S_t = Rep(X_t)$ is defined as a 49-dimensional vector summarizing the current feature space. It includes seven descriptive statistics (count, standard deviation, min, max, and first/second/third quartiles) computed column-wise and then aggregated row-wise, capturing both global and local characteristics.

- **4) Reward Function:** The reward is defined as the performance improvement of the downstream model after adding $f_t$:

$$R(t) = y_t - y_{t-1} \tag{13}$$

   where $y_t$ and $y_{t-1}$ denote the performance metrics before and after the transformation, respectively.

The agents are trained to maximize the cumulative reward by minimizing the mean squared error in the Bellman equation, thereby learning transformation policies that consistently benefit downstream performance.

---
**Algorithm 1** RL-based Data Collection Procedure
---
1: **Input:** Initial feature set $X_0$, operator set $\mathcal{O}$, downstream model
2: **for** each episode $i = 1$ to $N$ **do**
3:  Initialize feature space $X_0$ and state $S_0$
4:  **for** each step $t = 1$ to $T$ **do**
5:    Select features: $f_h \leftarrow \alpha_h(S_{t-1})$, $f_t \leftarrow \alpha_t(S_{t-1})$
6:    Select operator: $op \leftarrow \alpha_o(S_{t-1})$
7:    Generate new feature: $f_t = f_h \oplus_{op} f_t$
8:    Update feature set: $X_t = X_{t-1} \cup \{f_t\}$
9:    Evaluate performance: $y_t \leftarrow$ downstream model performance on $X_t$
10:    Compute reward: $R(t) = y_t - y_{t-1}$
11:    Update state $S_t \leftarrow Rep(X_t)$
12:    Update agents' policies using $R(t)$
13:  **end for**
14: **end for**
15: **Output:** Collected high-quality feature set
---

# D Experiments

## D.1 Datasets

To comprehensively evaluate the effectiveness and generalizability of our method, we conduct experiments on 14 benchmark datasets encompassing both classification and regression tasks. These datasets are sourced from widely used repositories, including UCIrvine, LibSVM, and OpenML. As summarized in Table 6, the datasets vary significantly in terms of sample size (ranging from 267 to 4601), feature dimensionality (from 19 to 10,936), and task type. This diversity allows us to assess model robustness across heterogeneous data distributions and application scenarios.

| Dataset | Source | Task | Samples | Features |
|---|---|---|---|---|
| SpectF | UCIrvine | C | 267 | 44 |
| SVMGuide3 | LibSVM | C | 1243 | 21 |
| German Credit | UCIrvine | C | 1000 | 24 |
| Messidor Features | UCIrvine | C | 1151 | 19 |
| SpamBase | UCIrvine | C | 4601 | 57 |
| Ap-omentum-ovary | OpenmlML | C | 275 | 10936 |
| Ionosphere | UCIrvine | C | 351 | 34 |
| Openml_586 | OpenmlML | R | 1000 | 25 |
| Openml_589 | OpenmlML | R | 1000 | 25 |
| Openml_607 | OpenmlML | R | 1000 | 50 |
| Openml_616 | OpenmlML | R | 500 | 50 |
| Openml_618 | OpenmlML | R | 1000 | 50 |
| Openml_620 | OpenmlML | R | 1000 | 25 |
| Openml_637 | OpenmlML | R | 500 | 50 |

Table 6: Dataset statistics.

## D.2 Baselines

We compare our method with 10 widely-used algorithms: 1) **RDG**, which randomly generates new feature transformations; 2) **PCA** [18], which generates new features based on linear correlations among original features; 3) **LDA** [3], which constructs a new feature space through matrix factorization; 4) **ERG**, which expands the feature space by applying selected operators to each feature, followed by feature selection; 5) **AFAT** [11], which repeatedly generates new features and applies multi-step feature selection; 6) **NFS** [4], which models the feature transformation process as a decision-making task and leverages reinforcement learning for optimization; 7) **TTG** [15], which formulates feature transformation as a graph search problem and employs reinforcement learning

to explore it; 8) **GRFG** [26], which builds a cascading agent architecture and proposes a feature group crossing strategy to enhance RL-based transformation search; 9) **MOAT** [27], which adopts a reinforcement learning-based data collector and explores the optimal feature set in the embedding space. 10) **ELLM** [6], which integrates evolutionary algorithm and large language model to iteratively generate high-quality feature sets.

### D.3  Experimental Environment

All experiments are conducted on the Ubuntu 22.04.3 LTS operating system, Intel(R) Core(TM) i9-13900KF CPU @ 3GHz, and 1 RTX 6000 Ada GPU with 48GB of RAM, using the framework of Python 3.11.4 and PyTorch 2.0.1.

### D.4  Details of Diffusion Model

Our latent diffusion model comprises 8 identical Transformer blocks. To accelerate training and improve convergence, we adopt the Min-SNR weighting strategy proposed in [9], which emphasizes learning from informative timesteps during the early training phase. The model is trained for 800 epochs using the standard latent diffusion loss as mentioned in [32]. During inference, we apply reward guidance with a reward scale set to 100 to enhance the quality of conditional generation. More design details and hyperparameters can be found in our publicly available code.

### D.5  Experimental Results

In this section, we present a more detailed ablation study. Each experiment is conducted five times, and the average performance is reported in Table 7.

Table 7: Performance of variant models and DIFFT on 8 datasets.

| Dataset | Variants | | | | DIFFT |
|---|---|---|---|---|---|
| | NAR | AR | NoR | CS | |
| SVMGuide3 | 0.840 | 0.856 | 0.831 | 0.830 | **0.871** |
| Messidor Features | 0.691 | 0.704 | 0.722 | 0.730 | **0.737** |
| SpectF | 0.792 | 0.823 | 0.851 | 0.876 | **0.881** |
| Ionosphere | 0.958 | 0.956 | 0.955 | 0.943 | **0.963** |
| Openml_586 | 0.621 | 0.628 | 0.623 | 0.610 | **0.633** |
| Openml_589 | 0.510 | 0.606 | 0.603 | 0.561 | **0.619** |
| Openml_616 | 0.230 | 0.302 | 0.237 | 0.289 | **0.324** |
| Openml_618 | 0.487 | 0.617 | 0.623 | 0.620 | **0.630** |

## E  Limitations

Although DIFFT delivers strong accuracy and robustness, two practical limitations remain. First, its task-optimal design requires regenerating a tailored feature set for every downstream model; when hundreds of learners or rapidly changing tasks are involved, the cumulative sampling and decoding cost can become non-trivial. Second, the current implementation has been validated on tables with up to roughly 10 K raw features; scaling to ultra-wide domains such as genome-scale assays or large-scale click logs would place increasing memory and time pressure on the VAE and the diffusion model. Future work will focus on extending DIFFT to a task-agnostic foundation model for feature generation.

