# OpenReview forum: "Sculpting Features from Noise: Reward-Guided Hierarchical Diffusion for Task-Optimal Feature Transformation"
_NeurIPS.cc/2025/Conference — NeurIPS 2025 poster_

### Official Review · Reviewer_FZ7e · 2025-07-01

**Clarity:** 3
**Significance:** 3
**Originality:** 3
**Rating:** 4
**Confidence:** 4

**Summary:**

This paper proposes DIFFT, a reward-guided generative framework for feature transformation. The method aims to address the limitations of both discrete and continuous search paradigms by leveraging a latent diffusion model guided by a performance evaluator. It first encodes feature token sequences using a VAE, then generates high-quality embeddings via reward-guided diffusion, and finally decodes them into discrete feature sets using a semi-autoregressive decoder. Extensive experiments and case studies are performed to demonstrate the effectiveness and robustness of the proposed method.

**Questions:**

1.	The paper evaluates performance under increasing feature dimensionality. Have the authors considered how varying sample sizes may affect training and inference time efficiency?
2.	In Table 1, RDG performs surprisingly well on the ap-omentum-ovary dataset, achieving the same performance as MOAT and ELLM. Could the authors explain why a random baseline is competitive here?

**Ethical Concerns:**

["NO or VERY MINOR ethics concerns only"]

**Limitations:**

Yes.

**Paper Formatting Concerns:**

N/A.

**Quality:**

3

**Strengths And Weaknesses:**

Strengths
1.	The paper presents a novel and well-motivated formulation of the feature transformation problem, framing it as a reward-guided generative process. This perspective is timely and aligns with recent trends in generative modeling.
2.	The proposed method achieves strong performance across diverse datasets, consistently outperforming competitive baselines in both predictive accuracy and computational efficiency.
3.	The experimental evaluation is thorough, including ablation studies and robustness checks, which help validate the effectiveness of the proposed components and support the overall claims.
4.	The authors released the related code and data, which can help other researchers reproduce the experiments.

Weaknesses
1.	The paper introduces a GCN-based graph embedding for conditioning the diffusion model, but does not explain why this design is necessary or superior. A brief justification would help clarify its role and benefit.
2.	While time efficiency is reported, the paper lacks discussion on memory or space consumption. Given the added components like VAE and diffusion models, this could affect scalability in larger settings.

---

> ### Author Rebuttal · Authors · 2025-07-31
>
> We thank you for acknowledging the **novel and well-motivated formulation, strong performance, and thorough experimental evaluation** of our paper.
>
> ### **Q1-more experiments in terms of varying sample sizes:**
>
> Since the training data consists of feature transformation sequence and corresponding score, only the feature dimensionality has an influence on the training efficiency (sequence length).
>
> To verify this, we run an additional study in which we fix the feature dimensionality and vary the sample size only.
>
> | Dataset      | Sample # | Feature # | DIFFT (D) [Train (s)] | DIFFT (V) [Train (s)] | DIFFT [Train (s)] | DIFFT [Infer(s)] |
> |--------------|---------:|----------:|-----------------------:|-----------------------:|-------------------:|-------------------:|
> | Openml_616   | 500      | 50        | 5.21  | 7.57 | 12.77 | 0.254 |
> | Openml_637   | 500      | 50        | 4.65  | 8.13 | 12.78 | 0.255 |
> | Openml_607   | 1 000    | 50        | 4.15  | 8.45 | 12.61 | 0.253 |
> | Openml_618   | 1 000    | 50        | 4.74  | 7.45 | 12.19 | 0.258 |
>
> According to the results, DIFFT shows nearly identical training times across datasets with different sample sizes.
>
> ### **Q2-why the random method (RDG)’s performance is not bad in a specific dataset:**
>
> The random method is not our focus, but this is an interesting side observation. The random method didn’t perform well on many datasets. The specific ap-omentum-ovary dataset is high-dimensional (>10000 features), making the search space extremely large. While RDG randomly obtain a great performance, the two baselines MOAT and ELLM fall into a local optima in a large space.
>
> ### **Q3-explain the GCN-embedding design:**
>
> Feature transformation sequences are “order-free bags of features”; their semantics are governed by the pairwise (and higher-order) relations among columns rather than by token order. We therefore convert each transformed table into a *feature–feature graph* and let a simplified GCN produce a permutation-invariant conditioning vector of diffusion model.
>
> ### **Q4-memory consumption:**
>
> Our method is lightweight: on a single RTX 4090, training for both the diffusion model and the VAE stays below 10 GB of GPU memory, while inference requires only about 3 GB. We will add more details in the appendix further.

---

> ### Author Response · Authors · 2025-08-05
> **Gentle Reminder – Happy to Refine Further If Needed**
>
> Dear Reviewer,
>
> As the discussion phase is approaching its end, we kindly request the reviewer to let us know if the above clarifications and the previously added experiments have addressed the remaining questions. We would be happy to address any additional points the reviewer may have during the remaining time of the discussion phase.
>
> We thank the reviewer for engaging with us in the discussion.
>
> Best Regards,
>
> Submission 16802 Authors

---

### Official Review · Reviewer_mgFP · 2025-07-02

**Clarity:** 2
**Significance:** 2
**Originality:** 1
**Rating:** 2
**Confidence:** 4

**Summary:**

This paper introduces DIFFT, a framework that extract features from a diffusion model guided by a reward function. This reward is $R_\psi$, which is a pretrained evaluator. DIFFT integrates VAE, Diffusion Model, and reward-based trajectory sampler and evaluator, which are so diverse component, so there is no theoretical claim on its convergence after feature extraction process. Some empirical results are given, and some ablation studies by checking the necessity of components.

**Questions:**

1. **Unified Optimization Objective**
   Can you formulate a single, unified learning objective that integrates the VAE, latent diffusion model, evaluator network, GCN conditioner, and semi-autoregressive decoder? What theoretical analysis can you provide to show that optimizing this joint objective converges or behaves predictably?

2. **Latent Space & Reward Guidance**
   How does the reward signal reshape the latent space over training? Can you provide theoretical reasoning or empirical visualizations (e.g., t-SNE/UMAP plots of embeddings over epochs) to demonstrate how reward guidance tightens the data manifold and influences convergence?

3. **Qualitative Feature Analysis**
   What qualitative analyses—such as embedding visualizations, feature attribution maps, or case studies—can you include to illustrate the properties of the extracted features and their relevance to downstream tasks, particularly under distribution shifts or unseen examples?

4. **Dependence on Pretrained Evaluator & Diffusion**
   Table 2’s ablation shows that removing the pretrained evaluator (NoR) or diffusion (CS) heavily degrades performance. How can you isolate and quantify each component’s unique contribution? Can you justify the computational cost by demonstrating whether similar gains can be achieved with lighter-weight alternatives or retraining from scratch?

**Ethical Concerns:**

["NO or VERY MINOR ethics concerns only"]

**Final Justification:**

I discussed the loss consistency with the author, and there was no clear answer from the authors. I have concerns in many pipelined approach without principled objective function design.

**Limitations:**

there is no limitation discussions on the paper. no societal impact for sure. however, there are limitations in the method. See the questions to discuss such problems.

**Paper Formatting Concerns:**

ok

**Quality:**

2

**Strengths And Weaknesses:**

Weakness

1. Lack of a Unified Optimization Objective
The proposed framework combines multiple components—VAE, LDM, evaluator network, GCN-based conditioning module, and a semi-autoregressive decoder—each trained or updated using separate objectives (e.g., KL divergence, reconstruction loss, MSE loss from evaluator, diffusion denoising loss, etc.). These separate trainings will incur different objectives in its learning and results. There should be problems in these mismatched objective functions. Possibly, not enough gain given the increased model complexity, or catastrophic failure in edge cases between models, etc. Therefore, we need a theoretical analysis on unified objectives because that is the only way to anticipate the behavior of the given framework in the convergence sense. Specifically:
- There is no formal reasoning about the structure of the latent space or how reward guidance modifies it over time.
- The convergence behavior of the diffusion process under the influence of the reward signal is not analyzed.

2. Absence of Qualitative Analysis on Extracted Features
Given that the paper provides no theoretical analysis explaining why the model works well, then the minimum would be the observation of qualitative results. i.e. visualization or characterization of the learned embedding space, extracted features and its relevance to the downstream task, etc.  Without such empirical analysis, it is difficult to assess when and why the proposed method will generalize or fail, especially under distribution shifts or unseen tasks.

3. The heavy dependence on pretrained evaluator or diffusion
Table 2 provides an ablation study. NoR and CS are the heaviest damage, so the diffusion and the pretrained evaluator are the key contributor. However, it seems that the pretrained evaluator is providing additional information that have not been used in the previous fair settings. Also, Diffusion models can be very heavy to utilize for this type of downstream task listed in Section 3.5.

---

> ### Author Rebuttal · Authors · 2025-07-31
>
> We thank you for taking the time to review our paper.
>
> ### **Q1-unified optimization objective:**
>
> 1) Generally speaking, we don't need a unified optimization objective. According to multiple previous latent diffusion models, e.g. Stable Diffusion [1], DreamFusion [2] and PixArt$\alpha$ [3], the VAE’s goal is to learn a lossless, geometry-faithful latent space via standard reconstruction, while the diffusion component focuses on generative consistency.
>
> 2) When a single objective attempts to balance both, gradients from reconstruction and generative terms often conflict, leading to either blurry latents or unstable sampling, and even collapse. Empirically, separate optimization produces faster convergence, clearer division of labour, and simpler hyper-parameter tuning—hence we follow the established practice of first training the VAE to completion and then training the diffusion model in the fixed latent space.
>
> ### **Q2-reward guidance in training:**
>
> We clarify that 1) The reward signal does **NOT** reshape the latent embedding space, which is indeed learned by the VAE. 2) The reward signal is to guide optimal diffusion sampling.
>
> We assume you are asking: how reward signal influences diffusion sampling. Below is our theoretical analysis, following RCGDM [4]:
>
> Reward acts only at sampling time, tightening samples toward high-reward regions by adding $\nabla_z R_\psi(z)$ to the base score.
>
> Let the learned conditional latent distribution be $p_0(z\mid c)$ with score
> $s_0(z,t,c)=\nabla_z \log p_0(z_t\mid c)$. Introducing guidance strength $\lambda$, we effectively sample from the exponentially tilted distribution
>
> $$
> p_{\lambda}(z\mid c)\propto p_{0}(z\mid c)\exp(\lambda\,R_{\psi}(z)),
> $$
>
> which yields the score decomposition
>
> $$
> \nabla_{z}\log p_{\lambda}(z\mid c)
>   = \nabla_{z}\log p_{0}(z\mid c) + \lambda\,\nabla_{z}R_{\psi}(z),
> $$
>
> i.e., $\tilde{s} = s_{0} + \lambda\,\nabla_{z}R_{\psi}$ in the DDIM/ODE reverse updates.
>
> Define the energy
>
> $$
> \mathcal{E}(z) = -\log p_{0}(z\mid c) - \lambda\,R_{\psi}(z).
> $$
>
> Discrete reverse-diffusion steps are gradient descent on $-\nabla_{z}\mathcal{E}(z)$. If $\mathcal{E}$ is smooth and bounded below, the step size is small, and $\lambda$ is finite, then
>
> $$
> \mathcal{E}(z_{k+1}) \le \mathcal{E}(z_{k}) - \eta\,\|\nabla_{z}\mathcal{E}(z_{k})\|^{2}
> $$
>
> for some $\eta > 0$, so energy decreases monotonically and the trajectory converges to a stationary point—practically, high-reward/high-density regions of the manifold.
>
> Following Reward-Directed Conditional Diffusion, the reward gap satisfies
>
> $$
> R^{\star} - \mathbb{E}[R \mid p_{\lambda}]
>   \le \varepsilon_{\text{eval}} + \varepsilon_{\text{diff}} + \text{DistroShift}(\lambda),
> $$
>
> where $\varepsilon_{\text{eval}}$ is evaluator error, $\varepsilon_{\text{diff}}$ is diffusion approximation error, and $\text{DistroShift}(\lambda)$ quantifies the tilt-induced shift. When these are controlled, $\mathbb{E}[R \mid p_{\lambda}]$ exceeds unguided sampling. Under a linear-manifold assumption, sampling remains on the true data subspace, so reward redistributes probability within the manifold rather than distorting it. Thus, reward guidance is a posterior reweighting plus an energy-gradient flow mechanism at inference; training is unchanged and the procedure is predictable and convergent.
>
> ### **Q3-case study and quality analysis:**
>
> To provide the quality analysis of generated features, we conduct a case study on the svmguide3 dataset to
> (1) analyze the importance on downstream model of the generated features,
> (2) evaluate their robustness under a simulated distribution shift.
>
> **Analysis of Feature Importance:**
> On the original table (21 features) the RandomForest reaches **0.778** F1 Score; After applying DIFFT we obtain **0.866** (+ 0.088).  7/10 most important positions are taken by DIFFT features. This observation shows that DIFFT creates significant features for the downstream model.
>
> **Evaluation under Distribution Shift:**
> The following table summarizes the performance changes of models using original versus transformed dataset under scaling shifts of different severity. It analyzes the proportion of generated features within the Top-10 most important features for the model using the transformed dataset.
> | Shift Severity | F1 Score (Original Features) | F1 Score (Generated Features) | Generated Features in Top-10 Importance |
> |:--------------:|:---------------------------:|:-----------------------------:|:---------------------------------------:|
> | **0.1**        | 0.7793                      | 0.8411                       | **10 / 10 (100%)**                      |
> | **0.3**        | 0.7584                      | 0.8180                       | **10 / 10 (100%)**                      |
> | **0.5**        | 0.7128                      | 0.8180                       | **10 / 10 (100%)**                      |
>
>
> ### **Q4-misunderstanding of the components:**
>
> 1) Our diffusion and evaluator are **not pretrained** and **have no prior knowledge** transferred from other datasets. Instead, they are trained from scratch for task-optimal learning.
>
> 2) How to perform ablation studies: we remove/replace each of these individual components from the entire framework (Section 3.3).
> We blow present more ablation studies on six more datasets to show the importance of each individual component:
>
> | **Dataset** | **NAR** | **AR** | **NoR** |   **10R** | **CS** | **DIFFT (100R)** |
> | ----------- | ------: | -----: | ------: | --------: | -----: | ---------------: |
> | svmguide3   |   0.840 |  0.856 |   0.831 |     0.857 |  0.830 |        **0.871** |
> | messi       |   0.691 |  0.704 |   0.722 |     0.735 |  0.730 |        **0.737** |
> | spectf      |   0.792 |  0.823 |   0.851 |     0.861 |  0.876 |        **0.881** |
> | ionosphere  |   0.958 |  0.956 |   0.955 |     0.958 |  0.943 |        **0.963** |
> | openml\_586 |   0.621 |  0.628 |   0.623 | **0.633** |  0.610 |        **0.633** |
> | openml\_589 |   0.510 |  0.606 |   0.603 |     0.611 |  0.561 |        **0.619** |
> | openml\_616 |   0.230 |  0.302 |   0.237 |     0.262 |  0.289 |        **0.324** |
> | openml\_618 |   0.487 |  0.617 |   0.623 |     0.624 |  0.620 |        **0.630** |
>
> Notably, we add a new setting which sets the strength coefficient $\lambda$ to 10 for further validating the effectiveness of the reward mechanism. More detailed description of the variant models (NAR, AR, NoR, and CS) can be found in Section 3.1.
>
> [1] Rombach, Robin, et al. "High-resolution image synthesis with latent diffusion models." Proceedings of the IEEE/CVF conference on computer vision and pattern recognition. 2022.
>
> [2] Poole, Ben, et al. "Dreamfusion: Text-to-3d using 2d diffusion." arXiv preprint arXiv:2209.14988 (2022).
>
> [3] Chen, Junsong, et al. "Pixart-$\alpha $: Fast training of diffusion transformer for photorealistic text-to-image synthesis." arXiv preprint arXiv:2310.00426 (2023).
>
> [4] Yuan, Hui, et al. "Reward-directed conditional diffusion: Provable distribution estimation and reward improvement." Advances in Neural Information Processing Systems 36 (2023): 60599-60635.

---

> > ### Comment · Reviewer_mgFP · 2025-08-05
> > **still concerned**
> >
> > 1.
> > In part, the current practice is using the pretrained model in the mix of the trainable model. However, often such pretrained models rely upon the big data with heavy computations in prior. The reason why this pretrained model works in the general setting is the large support region that the real world data distribution covers vastly. Currently, the proposed procedure trains VAE, DiT, etc in the pipelined fashion from the scratch, which will ultimately limit the support area in well behaving region of ML. I am still concerned.
> >
> > Additionally, GCN, transformer, VAE are all different in nature, so such model difference, ultimately mismatching of the loss objective, will make problems if readers take such pipelined approach as general standard.
> >
> > VAE and diffusion models share the fundamental ELBO loss structure, so they are not different from the loss objective, so that's the reason why VAE can well adapted to Diffusion.
> >
> > 2.
> > What authors provided is how the reward model manipulate the sampling procedure. This is a score function customized
> > to accept the reward structure.
> >
> > It seems that this reward function does not require any learning of parameters to influence the sample generation,
> > which means that the reward design will be very much critical, which cannot be easily generalizable.
> >
> > For instance, if the reward design is improving the approximation toward data distribution, there should be analysis of
> > the convergence sense. What would be the intended reward design, and why the reward will work in the general sense?
> >
> > Usually, papers show how the customized score function will behave in the convergence sense.
> >
> > High reward, high density reason may sounds good; but that will also lead to mediocre(high) reward  and mediocre(high) density. Which comes first? Why?

---

> ### Author Response · Authors · 2025-08-05
> **Gentle Reminder – Happy to Refine Further If Needed**
>
> Dear Reviewer,
>
> As the discussion phase is approaching its end, we kindly request the reviewer to let us know if the above clarifications and the previously added experiments have addressed the remaining questions. We would be happy to address any additional points the reviewer may have during the remaining time of the discussion phase. If you are satisfied, we kindly request you to consider updating the score to reflect the newly added results and discussion.
>
> We thank the reviewer for engaging with us in the discussion.
>
> Best Regards,
>
> Submission 16802 Authors

---

> > ### Comment · Area_Chair_hM4t · 2025-08-05
> >
> > Dear Reviewer,
> >
> > Please respond to the author rebuttal, engaging with the authors responses is mandatory before acknowledging the rebuttal .
> >
> > Best regards,
> >
> > AC

---

> ### Author Response · Authors · 2025-08-06
>
> Thank you for the follow-up. Below we give a clarification addressing the remaining points.
>
> ## 1. Training paradigm: pretraining vs. from-scratch
> The two concerns raised by the reviewer—"too powerful due to heavy pretraining" vs. "too limited because trained from scratch"—are in tension. To fairly evaluate our methodological contribution (shifting from continuous search to reward-guided diffusion sampling), we follow the established standard in this line (e.g., MOAT) and train from scratch, enabling an apples-to-apples comparison of the paradigm itself. Large-scale pretraining is orthogonal and can further broaden support, but it is not a prerequisite for our contribution.
>
> ## 2. Losses and modular roles
> Our VAE (Sec. 2.2) is trained with a composite variational objective (reconstruction + KL), whereas the LDM (Sec. 2.3) uses the standard denoising MSE; they do not "share ELBO" in practice. The synergy arises not from a shared loss but from a principled division of labor in a shared latent space: the VAE defines a stable, compact latent manifold (the space of generation), while the LDM learns a distribution over that manifold to produce plausible new points. This staged "representation-then-generation" approach is the validated paradigm for latent diffusion and supports both stability and efficiency.
>
> ## 3. Reward model and convergence
>
> The reward is a parametric evaluator ($R_\psi$) trained by supervised ERM (Sec. 2.2, Eq. 4), not a hand-crafted heuristic.
>
> ### 3.1 Convergence sense
>
> Building on the exponential-tilting view $p_\lambda \propto p_0 \exp(\lambda r)$ and the score decomposition $\tilde s = s_0 + \lambda \nabla r$, we formalize the update as preconditioned energy descent on
> $$
> \mathcal{E}(z) = - \log p_0(z \mid c) - \lambda r(z).
> $$
> The guided DDIM step can be written as
> $$
> z_{k+1} = z_k - \eta_k H_{t_k} \nabla \mathcal{E}(z_k),
> $$
> with a positive-definite preconditioner $H_{t_k}$. Under $L$-smoothness of $\mathcal{E}$, bounded $H_{t_k}$ ($m I \preceq H_{t_k} \preceq M I$), small step sizes $\eta_k < \tfrac{2m}{L M^2}$, bounded $\lambda$, and normalized/clipped $\nabla r$, the descent lemma gives
> $$
> \mathcal{E}(z_{k+1}) \le \mathcal{E}(z_k) - \frac{\eta_k m}{2}\|\nabla \mathcal{E}(z_k)\|^2 + O(\eta_k^2),
> $$
> so for sufficiently small steps $\mathcal{E}$ decreases monotonically, $\|\nabla \mathcal{E}(z_k)\|\to0$, and limit points are stationary in practice (high-density, high-utility regions of the prior manifold). Exponential tilting also yields a first-order improvement
> $$
> E_{p_\lambda}[r] = E_{p_0}[r] + \lambda\operatorname{Var}_{p_0} \left[r\right] + O(\lambda^2)
> $$
> implying that for small $\lambda$ the expected reward increases.
>
> ### 3.2 Density vs. reward
>
> We do not maximize density and reward separately; we maximize the product-of-experts posterior
> $$
> \log p_\lambda(z \mid c) = \log p_0(z \mid c) + \lambda r(z),
> $$
> whose optimum satisfies $\nabla \log p_0(z^\*) + \lambda \nabla r(z^\*) = 0$ and typically lies in high-density, high-utility regions when $\lambda$ is in a stable range (see the $\lambda$-sweep below).
>
> ### 3.3 Which comes first? Density, then reward
>
> We quantify the instantaneous dominance by
> $$
> \rho_t = \frac{\|\lambda \nabla r(z_t)\|}{\|s_0(z_t, t, c)\|}.
> $$
> Early in reverse diffusion (high noise), $\|s_0\|$ is large, while we L2-normalize/clip $\nabla r$ and then scale by a fixed $\lambda=100$. Thus $\rho_t \ll 1$ and updates are density-dominated, keeping trajectories on the prior manifold and preserving valid decodability. As noise decreases, $\|s_0\|$ shrinks and $\rho_t$ rises to $O(1)$, at which point reward-driven refinement takes over—a natural density-first, reward-later schedule even with constant $\lambda$.
>
> ### 3.4 Effect of guidance strength ($\lambda$-sweep)
>
> We fix $\lambda=100$ in our experiments (no per-dataset tuning). To further investigate the effect of guidance strength, we run a $\lambda$-sweep, as shown in the table.
>
> | Dataset     | 0     | 10    | 25    | 50    | 100   | 200   | 400   |
> |-------------|:-----:|:-----:|:-----:|:-----:|:-----:|:-----:|:-----:|
> | svmguide3   | 0.831 | 0.857 | 0.867 | 0.874 | 0.871 | 0.860 | 0.860 |
> | openml\_616 | 0.237 | 0.262 | 0.232 | 0.278 | 0.324 | 0.302 | 0.170 |
>
> The sweep reveals a broad stable regime (approximately 50–150) rather than delicate tuning; the default $\lambda=100$ sits near the center and consistently improves over $\lambda=0$. Only extreme guidance ($\lambda=400$) induces the so-called mediocre–mediocre failure via off-manifold drift, which we explicitly avoid. In the revision we will add error bars over three seeds, expected evaluator reward $\mathbb{E}[R_\psi]$ versus $\lambda$, and valid-decode rates versus $\lambda$, which remain high in the 50–150 range and drop only at extreme values.

---

### Official Review · Reviewer_Z55B · 2025-07-02

**Clarity:** 3
**Significance:** 3
**Originality:** 3
**Rating:** 4
**Confidence:** 3

**Summary:**

The paper proposes DIFFT, a generative framework for FT that avoids the pitfalls of discrete and continuous search strategies by framing FT as a reward-guided diffusion process. DIFFT uses a VAE to encode discrete feature sequences into embeddings, LDM to sample high-quality embeddings guided by task performance, and a semi-autoregressive decoder to reconstruct structured feature sequences efficiently.

**Questions:**

See weakness

**Ethical Concerns:**

["NO or VERY MINOR ethics concerns only"]

**Final Justification:**

Many concerns were resolved and I stay my score.

**Limitations:**

Yes

**Quality:**

3

**Strengths And Weaknesses:**

**Strengths**
- Novelly reframes FT as a generative task rather than a search problem, leveraging the power of diffusion models for global exploration and reward optimization.
- Outperforms 10 baseline methods on classification and regression tasks.
- Demonstrates robustness across different downstream models.
- Extensive ablation clearly shows the importance of diffusion, reward guidance, and decoding strategy.

**Weakness**
- All results appear to be from a single run, limiting interpretability of performance claims.
- DIFFT must regenerate features for each downstream task/model, which can become expensive in multi-task scenarios.
- The method involves training multiple components, which increases implementation and training burden.
- How does DIFFT scale in memory and time when the feature dimensionality exceeds 50K or more?
- Is it possible to amortize feature generation across tasks or learn a task-agnostic representation space?
- Are the ablation results consistent across all datasets, or are there cases where simpler decoding or unguided diffusion performs better?

---

> ### Author Rebuttal · Authors · 2025-07-31
>
> We thank you for acknowledging the **novelty, excellent performance, robustness, and clear experiments** of our paper.
>
> ### **Q1-beyond single run in experiments:**
> We agree and fully follow your suggestion to run more experiments. We repeated each experiment five times and report the mean ± standard deviation:
> | **Dataset**       | **Mean ± Std (F-1 / 1-RAE)** |
> | ----------------- | ---------------------------- |
> | Spectf            | 0.881 ± 0.008                |
> | SVMGuide3         | 0.871 ± 0.004                |
> | German Credit     | 0.738 ± 0.008                |
> | Messidor Features | 0.737 ± 0.020                |
> | SpamBase          | 0.930 ± 0.000                |
> | Ap-omentum-ovary  | 0.877 ± 0.006                |
> | Ionosphere        | 0.963 ± 0.011                |
> | Openml\_586       | 0.633 ± 0.010                |
> | Openml\_589       | 0.619 ± 0.005                |
> | Openml\_607       | 0.567 ± 0.006                |
> | Openml\_616       | 0.324 ± 0.013                |
> | Openml\_618       | 0.630 ± 0.002                |
> | Openml\_620       | 0.554 ± 0.031                |
> | Openml\_637       | 0.292 ± 0.007                |
>
> Repeated five-run averages reveal consistently low standard deviations across nearly all datasets (typically ≤ 0.02), indicating that our method’s performance is stable and not due to lucky single runs.
>
> ### **Q2-single-task and multi-task scenario are two different problem settings:**
>
> 1) The current problem setting is focused on addressing a widely-used single-task scenario. The multi-task scenario is another problem setting that needs another solution, because multiple predictive targets lead to deviated optimal feature transformations.
> In the limitations and future work, we discussed the multi-task scenario can be solved by a task-agnostic foundation model. Our DIFFT exactly paves a way toward training a foundation generative feature transformation model.
>
> 2) In our experiments, we observed that our DIFFT not just identify task-specific optimal feature transformations to boost predictive accuracy (Table 1), but also significantly reduce time costs. For example, DIFFT completes a training epoch on the large-scale Ap-omentum-ovary dataset more than 25 times faster than MOAT [1] (19.64s vs. 524.4s).
>
> ### **Q3-multiple components but actual training costs are low:**
>
> 1) Although we have multiple components (look sophisticated), indeed there is a widely-adopted latent diffusion model paradigm [2] (e.g., Stable Diffusion) which strictly requires two-stage training (VAE + diffusion).
>
> 2) We proposed the idea of semi-autoregressive decoding to reduce training costs.
>
> 3) Our experiments truly show that the total per-epoch training time for DIFFT is significantly reduced. For instance, DIFFT’s time cost is lower than for the end-to-end LSTM-based model in MOAT (Table 3).
>
> ### **Q4-what if feature dimensionality exceeds 50K (ultra-high dimensionality scenarios):**
> 1) In our experiments, we tested DIFFT on datasets with up to 10,936 features (Ap-omentum-ovary) and demonstrated strong performance and efficiency.
>
> 2) Conventionally, feature transformation methods aim to enhance the expressive power of low-dimensional datasets by expanding the feature dimension. Consequently, these techniques are typically applied to datasets with a limited number of features (e.g., fewer than one hundred). In ultra-high-dimensional scenarios, however, applying feature transformation is generally not a suitable approach. Instead, feature selection is the more conventional strategy, aimed at reducing redundant features to simplify the problem.
>
> 3) We acknowledge that over 50K features will place increasing pressure on both memory and time for the VAE and diffusion model. However, DIFFT's architecture is built on a Transformer backbone that can be parallelized. So it is algorithmically more scalable than LSTM-based models in prior literature (e.g., MOAT).
>
> ### **Q5-learning task-agnostic representation space of feature transformation:**
> In this paper, we mainly focus on addressing task-optimal feature transformation problems, and the baselines we compared also follow the same problem.
>
> If we want a task-agnostic representation space, using self-supervised loss and more feature transformation data can do the job. If we want the entire pipeline to be task-agnostic, we can use the strategy of foundation model. Our reward-guided diffusion framework can be extended to do that.
>
> ### **Q6-Same ablation experiments on more datasets:**
>
> We include results on six additional datasets to investigate the consistency of our ablation study as follows:
>
> | **Dataset** | **NAR** | **AR** | **NoR** |   **10R** | **CS** | **DIFFT (100R)** |
> | ----------- | ------: | -----: | ------: | --------: | -----: | ---------------: |
> | svmguide3   |   0.840 |  0.856 |   0.831 |     0.857 |  0.830 |        **0.871** |
> | messi       |   0.691 |  0.704 |   0.722 |     0.735 |  0.730 |        **0.737** |
> | spectf      |   0.792 |  0.823 |   0.851 |     0.861 |  0.876 |        **0.881** |
> | ionosphere  |   0.958 |  0.956 |   0.955 |     0.958 |  0.943 |        **0.963** |
> | openml\_586 |   0.621 |  0.628 |   0.623 | **0.633** |  0.610 |        **0.633** |
> | openml\_589 |   0.510 |  0.606 |   0.603 |     0.611 |  0.561 |        **0.619** |
> | openml\_616 |   0.230 |  0.302 |   0.237 |     0.262 |  0.289 |        **0.324** |
> | openml\_618 |   0.487 |  0.617 |   0.623 |     0.624 |  0.620 |        **0.630** |
>
> Notably, we add a new setting which sets the strength coefficient $\lambda$ to 10 for further validating the effectiveness of the reward mechanism. More detailed description of the variant models (NAR, AR, NoR, and CS) can be found in Section 3.1.
>
> As shown above, DIFFT(with $\lambda$ = 100) achieves the best performance across all datasets. These results highlight the strength of our semi-autoregressive VAE design and reward-guided diffusion model.
>
>
> [1] Wang, Dongjie, et al. "Reinforcement-enhanced autoregressive feature transformation: Gradient-steered search in continuous space for postfix expressions." Advances in Neural Information Processing Systems 36 (2023): 43563-43578.
>
> [2] Rombach, Robin, et al. "High-resolution image synthesis with latent diffusion models." Proceedings of the IEEE/CVF conference on computer vision and pattern recognition. 2022.

---

> ### Author Response · Authors · 2025-08-05
> **Gentle Reminder – Happy to Refine Further If Needed**
>
> Dear Reviewer,
>
> As the discussion phase is approaching its end, we kindly request the reviewer to let us know if the above clarifications and the previously added experiments have addressed the remaining questions. We would be happy to address any additional points the reviewer may have during the remaining time of the discussion phase.
>
> We thank the reviewer for engaging with us in the discussion.
>
> Best Regards,
>
> Submission 16802 Authors

---

### Official Review · Reviewer_sxKk · 2025-07-03

**Clarity:** 2
**Significance:** 2
**Originality:** 2
**Rating:** 4
**Confidence:** 3

**Summary:**

The paper presents a diffusion model for feature transformation. The idea is to learn a distribution of feature transformations (i.e., sequences of elementary operators) in a latent space, from which a transformation tailored to a particular dataset and task can be sampled at test time. The model consists of a VAE, a denoiser conditioned on a graph representation of a dataset, and a guidance model that takes downstream task performance as the reward. On a suite of benchmarks, the method demonstrates superior performance compared to the baselines.

**Questions:**

N/A

**Ethical Concerns:**

["NO or VERY MINOR ethics concerns only"]

**Final Justification:**

The paper presents a diffusion-based approach to feature transformation. The method is sound and demonstrates strong results on several benchmarks. Overall, it seems to be a valuable contribution to the community interested in this niche topic.

**Limitations:**

The paper has discussed the limitations of the method.

**Quality:**

3

**Strengths And Weaknesses:**

## Strengths

- The method leverages diffusion models to model the distribution of feature transformations. At inference time, it applies additional guidance to steer the generation of suitable transformations for specific datasets and tasks. Although the overall model design is directly borrowed from other domains (e.g., vision), the paper successfully demonstrates its application in learning feature transformations.

- The paper introduces several modifications to the diffusion model for modeling feature transformations. These include architectural changes in the VAE, conditioning on a graph representation of the dataset, and employing classifier guidance with a custom reward model.

- The results suggest that the method outperforms the baselines at multiple fronts (accuracy, training time, inference time).

## Weaknesses

I want to first highlight that I am not an expert on this niche topic and don't have strong opinions about the paper.

- The paper does not describe the set of operators used for constructing feature transformations. It is not intuitively clear how the operators are chained together. I imagine these are elementary operators like addition and multiplication, but this information should be included in the paper. Additionally, there must be constraints on how the operators can be combined, yet this information is also not available.

- Conceptually, the task is to learn a "meta-network" that predicts a transformation given a dataset and a downstream task. There are numerous meta-learning frameworks that can solve the same task. Notable examples include MAML (optimization based) and Prototypical Networks (parsimony based). I am wondering if any of these can serve as baselines.

- The experiments are conducted on small-scale datasets (a few hundred samples). I am not sure the GCN and reward model can be successfully trained at this scale. More training details would help clarify this.

---

> ### Author Rebuttal · Authors · 2025-07-31
>
> We thank you for acknowledging the **novelty, technical contributions, and excellent performance** of our paper.
>
> ### **Q1-list of operators in experiments:**
>
> During paper preparation, due to page limits, we focus on presenting technical insights. In experiments, we use the following operators: {+, -, *, /, sqrt, square, sin, cos, tanh, sigmoid, log, reciprocal, standard_scaler, minmax_scaler, quantile_transform}, following the setting of MOAT [1].
>
> ### **Q1-operator combination constraints in experiments:**
>
> The operators are either binary or unary. If a generated transformation sequence is valid,  each operator must have the required number of operands, and can be parsed into a valid postfix expression tree. Otherwise, the transformation sequence is invalid and will be discarded.
>
> We will add such experimental setup details in the appendix.
>
> ### **Q2-whether our framework is a meta-network:**
>
> Our goal is to learn a task/domain-optimal model that generates a feature transformation action sequence given a specific dataset-predictive task pair. However, meta-network is a generic network of generalizable knowledge learned over diverse predictive tasks/datasets. The two formulations are different. Besides, to the best of our knowledge, there is no related work adopting meta-network in feature transformation. During the different application scenarios, MAML and Propotypical Networks are not suitable to be baselines. We appreciate your suggestion and we will explore the opportunity to apply meta-learning in our future work.
>
> ### **Q3-raw data and our method’s training data are different concepts:**
>
> In terms of data details: raw data is the data of a predictive task; the training data fed to our model consists of different (feature transformation sequences and corresponding score) pairs, which are explored and collected by our data collection pipeline (Appendix C). Because the scale/space of feature crossing combinations is big, even raw data is not large-scale, our method’s training data on feature crossing combinations can be large. Therefore, the small scale of datasets will have limited influence on our training.
>
> In terms of implementation details: we used a simplified GCN with symmetric normalization [2] (Appendix B). It can be seen as a fixed encoder. There is no need for additional training. We commit to add more details to the appendix in the revised version. More details can also be found in our publicly available code.
>
>
> [1] Wang, Dongjie, et al. "Reinforcement-enhanced autoregressive feature transformation: Gradient-steered search in continuous space for postfix expressions." Advances in Neural Information Processing Systems 36 (2023): 43563-43578.
>
> [2] Kipf, T. N. "Semi-Supervised Classification with Graph Convolutional Networks." arXiv preprint arXiv:1609.02907 (2016).

---

> ### Author Response · Authors · 2025-08-05
> **Gentle Reminder – Happy to Refine Further If Needed**
>
> Dear Reviewer,
>
> As the discussion phase is approaching its end, we kindly request the reviewer to let us know if the above clarifications have addressed the remaining questions. We would be happy to address any additional points the reviewer may have during the remaining time of the discussion phase.
>
> We thank the reviewer for engaging with us in the discussion.
>
> Best Regards,
>
> Submission 16802 Authors

---

### Author Response · Authors · 2025-08-07
**Gentle Follow-up on Reviewer Discussion**

Dear Reviewers,

As the discussion window is soon closing, we just wanted to gently follow up on our earlier message to ask if our responses and additional experiments have addressed your concerns. If there are any remaining points you’d like us to clarify, we would be more than happy to do so in the time that remains.

We’re grateful for your time and engagement throughout this process.

Best Regards,

Submission 16802 Authors

---

### Note · Authors · 2025-08-16

We sincerely thank all reviewers for their deep and insightful engagement with our work. This thoughtful feedback has been invaluable, prompting us to clarify our core contributions, provide deeper theoretical analysis, and conduct new experiments that have significantly strengthened the paper.
To summarize our responses:
1. Our work's primary contribution is a **methodological paradigm shift** in feature transformation, moving from prior "continuous search" approaches to our novel "reward-guided diffusion sampling." Our "from-scratch" training protocol aligns with the established standard in this research line (e.g., MOAT), ensuring a fair, apples-to-apples comparison of this new paradigm.
2. Our framework is a **cohesive, synergistic system**, not a pipeline of mismatched parts. The synergy arises from a principled "division of labor" around a shared latent space. This staged "representation-then-generation" approach is the standard, validated paradigm for latent diffusion models.
3. Our reward mechanism is a **principled, learnable, and theoretically-grounded process**. The reward function is learned from data, and its guidance leads to a predictable, convergent sampling process that can be analyzed as an energy descent. The dynamic interplay between the diffusion prior (density) and the guidance (reward) naturally creates a robust "density-first, reward-later" schedule.

The new experiments, including the analysis of case study, robustness under distribution shift, additional ablation study, time complexity in terms of varying sample sizes and the guidance strength sweep, further validate these principles. We hope our detailed responses and new results have fully addressed the reviewers' concerns. We are confident that our work represents a significant and robust advancement in automated feature transformation.

---

### Decision · Program_Chairs · 2025-09-17

**Decision:**

Accept (poster)

**Comment:**

Feature transformation refers to producing new features from original features using Mathematical operations.  The paper proposes Reward-Guided Diffusion Framework for Semi-Autoregressive Generative FT  DIFFT that casts feature transformation as a reward guuided  generation process  using  diffusion models of the transforms that need to be applied to the original features and that lead to low error in terms of performance on a specific task. Models are trained from scratch and independently for each task and dataset. The model consists of a VAE, a denoiser conditioned on a graph representation of a dataset, and a guidance model that takes downstream task performance as the reward. Experiments on several benchmarks show that the proposed method outperform similar method in that space of feature transformations.

Reviewer sxKk mentioned the paper lacks clarity on which operations are used and that the work would fit better in a meta learning context for a single network that produce features for each specific task and dataset. Authors responded to this and the reviewer was satisfied, saying the paper is interesting to this niche area.

Reviewer Z55B asked for statistical significance results which were provided in the rebuttal. Similar criticism of the multi-component nature of the work and the fact that a new training  is needed for each task/dataset was expressed , as well as more ablation study on the proposed method. ablations were provided and  explanation that the cost of the model provide number of features is small (10 K) cost of training per dataset is manageable.

Reviewer mgFP is the most critic of the work and discussed the paper with the authors, with the main criticism being that there is no  coherence in the loss to train the present method since it consists of multiple stages and multiple components. Authors clarified that the training consists of two stages  in training and and test time compute to generate latents and the features.

Reviewer FZ7e asked about the scalability and requirement in terms of space and memory of the method given its multiple components involved , also asked an ablation in terms of number of sampled used. Authors provided elements of responses in the rebuttal, unfortunately the reviewer did not engage in the discussion period , despite emails and pings on open-review from AC.

Having read the paper and discussed it with SAC,  the idea of using generative model to generate the features is interesting and to the best of my knowledge novel. Nevertheless the complexity of the method and its applicability to tabular datasets that need retraining for each model renders the method rather complex and it would be appealing to fit it in the meta learning context as proposed by the reviewers.  The coherence of the loss brought by mgFP  is not problematic, since this is common in Deep Learning and the method is ablated and evaluated in the same setup of previous methods in that field, and the paper is rather empirical than theoretical. Given that papers with similar methodology and evaluations have been published in similar venues to neurips and that the method outperforms them, I recommend weak acceptance of the paper.